# Bioinformatics-Based Analysis of Key Genes in Steroid-Induced Osteonecrosis of the Femoral Head That Are Associated with Copper Metabolism

**DOI:** 10.3390/biomedicines11030873

**Published:** 2023-03-13

**Authors:** Baochuang Qi, Chuan Li, Xingbo Cai, Luqiao Pu, Minzheng Guo, Zhifang Tang, Pengfei Bu, Yongqing Xu

**Affiliations:** 1Graduate School, Kunming Medical University, No.1168, Chunrong West Road, Yuhua Street, Chenggong District, Kunming 650500, China; qibaochuang@126.com (B.Q.); lichuankaka@163.com (C.L.); caixingbo@yet.net (X.C.); guomz1412@163.com (M.G.); 2Department of Orthopaedics, 920th Hospital of the Joint Logistics Support Force of the Chinese People’s Liberation Army, Kunming 650032, China; nydbpf@126.com; 3Medical College, Dali University, Dali Bai Autonomous Prefecture, Dali 671003, China; tangzhifangtzf@163.com

**Keywords:** steroid-induced osteonecrosis of the femoral head, copper metabolism, gene expression omnibus, quantitative real-time polymerase chain reaction, bioinformatic

## Abstract

Osteonecrosis of the femoral head (ONFH) is a common disabling disease. Copper has positive effects on cells that regulate bone metabolism. However, the relationship between copper metabolism (CM) and steroid-induced ONFH (SONFH) remains unclear. The GSE123568 dataset was downloaded from the Gene Expression Omnibus. The differentially expressed CM-related SONFH genes (DE-CMR-SONFHGs) were identified via differential analysis and weighted gene coexpression network analysis (WGCNA). Receiver operating characteristic (ROC) analysis was performed for the predictive accuracy of key genes. Targeting drugs and the copper death-related genes (CDRGs) relevant to key genes were investigated. The bioinformatics results were confirmed via quantitative real-time polymerase chain reaction (qRT–PCR) and Western blot (WB) analysis. Two out of 106 DE-CMR-SONFHGs were identified as key genes (PNP and SLC2A1), which had diagnostic value in distinguishing SONFH from control samples and were related to various immune cell infiltrations. Eleven PMP-targeting drugs and five SLC2A1-targeting drugs were identified. The qRT–PCR, as well as WB, results confirmed the downregulation PNP and SLC2A1 and high expression of the CDRGs DLD, PDHB, and MTF1, which are closely related to these two key genes. In conclusion, PNP and SLC2A1 were identified as key genes related to SONFH and may provide insights for SONFH treatment.

## 1. Introduction

Osteonecrosis of the femoral head (ONFH) is a common disabling disease [1]. Approximately 150,000–200,000 new cases per year are reported in China [2], and approximately 20,000 to 30,000 new patients per year are diagnosed in the USA [3]. Nontraumatic ONFH is mainly the result of glucocorticoid use and chronic alcohol consumption [4]. Significant osteonecrosis results in the collapse of the articular cartilage of the femoral head, followed by early osteoarthritis (OA) of the hip. In clinical practice, individuals who are affected by steroid-induced osteonecrosis of the femoral head (SONFH) are usually young and middle-aged individuals [5]; the exact pathogenesis is still unclear, and effective prevention and early treatment options are scarce. Hence, it is vital to investigate the exact pathogenesis of SONFH and to discover ideal methods for the early diagnosis and treatment of SONFH.

Copper is a double-edged sword in cells. Copper is an indispensable co-factor for all organisms, and active homeostatic mechanisms work across copper concentration gradients to maintain very low intracellular copper concentrations in order to avoid the intracellular accumulation of free copper that can be hazardous to cells. Copper levels in mammals are tightly controlled through intracellular or systemic homeostatic mechanisms. However, excessive levels of intracellular copper ions cause damage because these copper ions produce free radicals and induce oxidative stress [6]. Copper metabolism includes copper uptake, distribution, sequestration, and excretion at the cellular and systemic levels in multicellular organisms. Excessive copper intake can cause undesired modulation of the immune response [7]. Chen et al. [8] showed that prolonged exposure of cells and tissues to excessive copper levels could activate p53-dependent or p53-independent pathways that lead to “programmed cell death” or “apoptosis”. Copper-induced “apoptosis” has been verified in spleen cells, thymocytes, and hepatocytes [9,10]. Copper (Cu) ions stabilize the expression of hypoxia-inducible factor-1α (HIF-1α) and upregulate vascular endothelial growth factor (VEGF) expression; high VEGF expression induces neovascularization to further promote bone development [11,12]. However, it has also been demonstrated that higher concentrations of copper significantly reduce the value-added of osteogenic precursor cells and decrease new bone formation [13]. Physically, steroid hormone application can lead to a disturbance in copper metabolism, causing changes in serum copper levels [14,15]. However, the relationship between disorders of copper metabolism and SONFH has not yet been revealed.

With the development of bioinformatics in recent years, microarray analysis using high-throughput platforms has been applied as effective means in exploring the molecular mechanisms of disease and identifying biomarkers [16]. Based on the microarray data in GSE123568 datasets, as did the previous literature [17,18], this study mainly focused on identifying the key genes in SONFH from the perspective of copper metabolism on the basis of bioinformatics and supplemented by the experimental verification of gene expression; this study aimed to provide new ideas for the treatment and prevention of SONFH.

## 2. Materials and Methods

### 2.1. Data Source

The SONFH-related dataset GSE123568 (30 SONFH blood samples and 10 control blood samples) was extracted from the Gene Expression Omnibus (GEO) database (https://www.ncbi.nlm.nih.gov/, accessed on 10 May 2022). Furthermore, 2062 copper metabolism-related genes (CMRGs) were derived from the GeneCards database (https://www.genecards.org/, accessed on 10 May 2022) with the search term “Copper metabolism”.

### 2.2. Differential Gene Expression Analysis

The differential gene expression analysis between SONFH samples and control samples in the GSE123568 dataset was performed using the R package “limma” [19]. Multiple testing correction was implemented using the method described by Benjamini and Hochberg, and criteria for identifying the differentially expressed SONFH genes (DE-SONFHGs) were an adjusted *p* value < 0.05 and |log_2_fold change (FC)| > 0.5.

### 2.3. Weighted Gene Coexpression Network Analysis (WGCNA)

To identify genes of which expression is highly correlated with SONFH in the GSE123568 dataset, WGCNA [20] was performed. The samples were clustered to determine the overall correlation of all samples and to exclude outliers in order to ensure the accuracy of the analysis. To ensure that the interactions between genes fit the scale-free distribution to the greatest extent, a soft threshold was determined. The minimal number of genes in each module was fixed at 30. The modules that positively and negatively correlated with SONFH were selected as key modules. Then, the module members (MMs) and gene significance (GS) were estimated, and a scatter plot was generated to identify the key module genes according to the criteria |MM| > 0.8 and |GS| > 0.2.

The DE-SONFHGs, CMRGs, and the key module genes that were identified were intersected with the jvenn tool to identify the differentially expressed CM-related SONFH genes (DE-CMR-SONFHGs).

### 2.4. Functional Enrichment Analysis of DE-CMR-ONFHGs

The DAVID database (https://david-d.ncifcrf.gov/, accessed on 10 May 2022) [21] was utilized to conduct Gene Ontology (GO) and Kyoto Encyclopedia of Genes and Genomes (KEGG) pathway enrichment analyses on the DE-CMR-SONFHGs [22,23,24]. A significance threshold of *p <* 0.05 and a number of enrichments (counts) of at least 2 were considered to indicate significantly enriched results.

### 2.5. PPI Network Construction

To investigate the interactions among DE-CMR-SONFHGs, a protein–protein interaction (PPI) network was developed with the Search Tool for the Retrieval of Interacting Genes (STRING) database. The confidence level was 0.4, and discrete proteins were eliminated to identify interaction relationship pairs.

### 2.6. Screening for Key Genes using a Machine Learning Algorithm

Least-Absolute Shrinkage and Selection Operator (LASSO) logistic regression and Support Vector Machine (SVM) algorithms were used to screen the key DE-CMR-SONFHGs. The R software “glmnet” package (version 4.0-2) [25] was used to perform 10-fold cross-validation, and the error rate was calculated under different features to select strongly correlated genes. Similarly, the SVM algorithm in “e1071” (version 1.7-9) [26] was utilized to sort the DE-CMR-SONFHGs. The recursive feature elimination (RFE) method was used to determine the importance and importance ranking of each gene, and the error rate and accuracy rate were calculated. The lowest point of the error rate was selected as the best combination, and the corresponding genes were considered candidate genes. The genes that intersected with the candidate genes that were identified via the LASSO analysis and SVM analysis were obtained using the jVenn tool. Additionally, to further explore the diagnostic value of key genes in the discrimination of SONFH samples and control samples, the R package “pROC” [27] was used to plot the receiver operating characteristic (ROC) curves of key genes in the GSE123568 dataset [28].

### 2.7. Single-Gene Gene Set Enrichment Analysis (GSEA)

To study the molecular mechanisms associated with the diagnostic key genes, GSEA software (V4.0.3) [29] was utilized to perform single-gene GSEA [30]. When the parameter “c2.cp.kegg.v7.4.symbols.gmt” was set, the KEGG pathway gene set was used as the enrichment background [22,23,24]; when the parameter “c5.go.bp.v7.4.symbols.Gmt” was set, the GO biological process gene set was set as the enrichment background; when the parameter “Phenopyte labels: Use a gene as the phenotype” was used, the expression values of two key genes were considered the phenotype file. Then, all other genes, separately in the gene sets, were used to calculate the correlation with each key gene. Arranged according to the correlation coefficient from high to low, these genes were used as the new gene sets to be tested. The enrichment of the GO and KEGG terms in the tested gene sets was examined, and the significant enrichment threshold was set as an NOM *p* value < 0.05.

### 2.8. Immune Cell Infiltration Analysis

The single-sample Gene Set Enrichment Analysis (ssGSEA) algorithm was used to analyze the abundance of 28 infiltrating immune cell populations in all samples in the GSE123568 dataset. Then, the R package “ggplot2” [31] was used to reveal the immune cell populations with differential abundance between SONFH samples and control samples using the Wilcoxon test method. The correlation between differential immune cell infiltration was calculated with “corrplot”. The correlation between key genes and differential immune cell infiltration was calculated via Spearman analysis.

### 2.9. Drug Predictive Analysis

According to the previous literature, I drugs that target the key genes were identified with The Drug Gene Interaction Database (DGIdb; www.dgidb.org, accessed on 10 May 2022), and Cytoscape software [32] was used to build a drug targeting network.

### 2.10. Analysis of the Relevance of Copper Death-Related Genes and Differentially Expressed Genes in Disease

Based on the 10 copper death-related genes (FDX1, LIPT1, LIAS, DLD, PDHA1, DLAT, PDHB, GLS, MTF1, and CDKN2A) that have been reported in the literature [33,34], the expression of 2 key genes (PNP and SLC2A1) and 10 copper death-related genes were extracted from the GSE123568 dataset, and the Pearson correlation coefficient between the key genes and the copper death-related genes was calculated.

Moreover, the expression of 10 copper death-related genes was extracted from the GSE123568 dataset and combined with the grouping information of the samples, and the R package “ggplot2” was used with the Wilcox test method to generate graphs showing copper death-related gene expression in the disease and control samples.

### 2.11. Verification of the Expression of Key Genes in Clinical Samples

Human samples were acquired from the Department of Orthopaedics of the 920th Hospital of the People’s Liberation Army Joint Security Force. A total of 13 peripheral blood samples were collected, 6 of which were collected from SONFH patients and 7 were collected from healthy participants [35]. Eight bone tissue samples (including 4 hormonal osteonecrosis and 4 normal femoral head bone tissue) were obtained from femoral head removed during surgery [36]. The Ethics Committee of the 920th Hospital of the Chinese Pe’ple’s Liberation Army Joint Security Force approved this study, and the individuals who participated in the study provided written informed consent.

Preoperatively, 3 mL of fasting peripheral anticoagulated blood was collected, and lymphocytes were separated using lymphocyte isolation solution. Total RNA was extracted with the TRIzol reagent (Invitrogen, Carlsbad, CA, USA) following the protocol. cDNA synthesis was performed using a reverse transcription kit (Takara, Tokyo, Japan) according to the manufacturer’s instructions. The thermal procedure of polymerase chain reaction (PCR) used in this study was initial denaturation at 95 °C for 1 min, denaturation at 95 °C for 20 s, annealing at 55 °C for 20 s, and extension at 72 °C for 30 s. The PCR was performed for 40 cycles. GAPDH was used as the positive control for this reaction, and each sample was calculated using the comparative Ct method (Table 1).

Bone tissue samples (including 4 hormonal osteonecrosis and 4 normal femoral head bone tissue) were collected. An appropriate amount of RIPA cracking solution was added, and the tissues was cracked on ice for 30 min. Total protein was conducted using a BCA protein concentration assay kit (Biyuntian, Shanghai, China) according to regulations. Samples with equal amounts of proteins were separated by performing electrophoresis and were probed with primary antibodies against DLD, PDHB, MTF1, SLC2A1, and PNP (Affinity or Proteintech). Membranes were then incubated with a horseradish peroxidase-conjugated secondary antibody, and proteins were visualized with enhanced chemiluminescence reagents. The target protein abundance was normalized to actin.

### 2.12. Statistical Analysis

All bioinformatics analyses were performed in R language. The Wilcoxon test was used to compare the data from different groups. The qRT–PCR and WB data were analyzed using the 2^−△△Ct^ method. Student’s t test was utilized to compare the differences in the RT–qPCR and WB data. If not otherwise stated, a *p* value smaller than 0.05 indicates significance.

## 3. Results

### 3.1. Screening of Differentially Expressed Genes

The workflow diagram (Figure 1) clearly shows the major steps of this paper. A total of 2036 DE-SONFHGs between the SONFH samples and control samples were identified, and these DE-SONFHGs included 1383 upregulated and 653 downregulated genes. The distribution of these DE-SONFHGs is presented Figure 2.

### 3.2. WGCNA

The overall clustering of the dataset samples was good, so no samples were removed (Appendix A). Then, the traits of the samples were classified, and sample clusters and clinical feature heatmaps were generated (Appendix A). According to the position of the blue line, the power threshold was determined to be 24 (R^2^ = 0.85), and 48 modules were obtained (Figure 3A,B and Appendix A). Subsequently, similar modules were analyzed and merged with the dynamic cutting tree algorithm with a MEDissThres equal to 0.2, and 16 modules remained after merging (Figure 3C). The module with the strongest positive correlation (bisque4 module, cor = 0.67, *p* = 3 × 10^−6^) and the module with the strongest negative correlation (dark-green module, cor = −0.83, *p* = 5 × 10^−11^) were selected as key modules; the bisque4 module included 2382 genes, and the dark-green module included 245 genes (Figure 3C). Furthermore, 896 hub genes were authenticated in the bisque4 module, and 145 hub genes were authenticated in the dark-green module (Figure 3D). Finally, a grand total of 1041 hub genes were identified. Moreover, 106 DE-CMR-SONFHGs were identified by taking the intersection of 1041 hub genes, 2036 DE-SONFHGs and 2062 CMRGs (Figure 3E).

### 3.3. Functional Enrichment Analysis of DE-CMR-ONFHGs

The 106 DE-CMR-SONFHGs were enriched in 143 GO-Biological Process (GO-BP), 50 GO-Cellular Component (GO-CC), and 34 GO-Molecular Function (GO-MF) terms, and 51 KEGG pathways (Figure 4A,B). These genes were mainly enriched in various GO terms related to protein binding, the membrane, and stimulus response, such as identical protein binding, plasma membrane, and inflammatory response. Furthermore, these DE-CMR-SONFHGs were mainly enriched in phagosomes, the PI3K-Akt signaling pathway, and neutrophil extracellular trap formation.

### 3.4. Linkage between Proteins

A PPI network was constructed to explore the interactions of 106 DE-CMR-SONFHGs using the STRING website; 387 protein interaction pairs, including 92 nodes, were identified (Figure 5). TLR4 had the greatest connectivity in the PPI network. In addition, 15 downregulated genes were included in the PPI network.

### 3.5. Screening for Key Genes using a Machine Learning Algorithm

LASSO logistic regression and the SVM algorithm were utilized to identify key DE-CMR-SONFHGs. When lambda.min was 0.1073883, the error rate was the lowest, and two candidate genes, namely PNP and SLC2A1, were identified via LASSO (Figure 6A,B). Additionally, eight candidate genes, including RNASET2, PNP, SLC2A1, REXO2, CYBA, SOAT1, TFDP1, and LYZ, were selected using SVM (Figure 6C and Table 2). Finally, two genes (PNP and SLC2A1) were identified in the results of both LASSO and SVM (Figure 6D). The ROC curves revealed that the area under the curve (AUC) values of PNP and SLC2A1 in the GSE123568 dataset were both above 0.9, indicating that each key gene had diagnostic value in distinguishing between SONFH samples and control samples (Figure 7).

### 3.6. Single Gene Enrichment Analysis

To investigate the underlying functions of PNP and SLC2A1, GSEA was performed. PNP was mainly enriched in “positive regulation of smoothened signaling pathway”, “positive regulation of cytokine production involved in immune response”, “Glycosphingolipid biosynthesis lacto and neolacto series”, and “B-cell receptor signaling pathway” (Figure 8A,B and Appendix A). SLC2A1 was mainly enriched in “cgmp metabolic process”, “vacuolar acidification”, and “glycosphingolipid biosynthesis lacto and neolacto series” (Figure 8C,D and Appendix A).

### 3.7. Immune Cell Infiltration Analysis

To explore the abundance of 28 infiltrating immune cell populations in all the samples, ssGSEA was performed between the SONFH and control samples. Twenty immune cell populations, such as MDSCs, eosinophils, and CD56bright NK cells, were significantly different between the SONFH and control samples (Appendix A). Among these differentially abundant cell populations, the abundance of immature dendritic cells was strongly negatively correlated with the abundance of activated B cells, and the abundance of CD56bright natural killer cells was strongly positively correlated with the abundance of activated B cells (Appendix A). PNP expression had a significant and the strongest negative association with the abundance of plasmacytoid dendritic cells, and PNP expression had a significant and the strongest positive association with the abundance of CD56bright natural killer cells (Appendix A). SLC2A1 expression had a significant and the strongest negative association with the abundance of T follicular helper cells, and SLC2A1 expression had the strongest positive association with the abundance of T helper 17 cells (Appendix A).

### 3.8. Drug Prediction Analysis

After the drug prediction analysis, 16 drugs in the DGIdb database were predicted to target PNP and SLC2A1; of these drugs, 11 drugs were predicted to target SLC2A1, and five drugs were predicted to target PNP (Appendix A).

### 3.9. Relevance of Copper Death-Related Genes and Differentially Expressed Genes to Disease

To investigate the correlation between copper death-related genes and differentially expressed genes in femoral head necrosis, we calculated correlations based on the expression of the two key genes that were identified and the expression of 10 copper death-related genes, and we determined the corresponding *p* values and correlation coefficients, r. Significantly correlated relationship pairs were identified based on the correlation threshold *p* value < 0.05 and |r| > 0.3 (Appendix A). The correlation results were visualized through the R package ‘ggplot2’ (Figure 9) according to the correlation ranking, where a significant and the strongest positive correlation was observed among SLC2A1, PNP, and CDKN2A, and a significant and the strongest negative correlation was observed among SLC2A1, PNP, and PDHB.

The expression of the 10 copper death-related genes was extracted from the GSE123568 dataset and combined with the sample grouping information. The expression of copper death-related genes in the disease samples and control samples was evaluated via the wilcox.test method (Figure 10). The results showed that four copper death-related genes (LIPT1, DLD, PDHB, and MTF1) were significantly different and were differentially upregulated in the disease samples.

### 3.10. Verification of the Expression of Key Targets through qRT–PCR and WB

qRT–PCR was performed on human blood samples to analyze the expression of PNP and SLC2A1. Compared to that in the normal group, the expression of PNP (*p =* 0.0362) and SLC2A1 (*p =* 0.0094) was notably reduced in the SONFH group (Figure 11A).

Compared to that in the normal group, the expression of LIPT1 (*p =* 0.4129) showed no significant difference, but DLD (*p* < 0.0001), PDHB (*p =* 0.0149), and MTF1 (*p =* 0.0326) was increased in the SONFH group (Figure 11A).

The results of WB were similar to those of qRT–PCR. Compared to that in the normal group, the expression of PNP (*p =* 0.0005) and SLC2A1 (*p =* 0.0025) was notably reduced in the SONFH group (Figure 11B,C).

Compared to that in the normal group, the expressions of DLD (*p =* 0.0279), PDHB (*p =* 0.0391), and MTF1 (*p =* 0.0009) was notably increased in the SONFH group (Figure 11B,C).

## 4. Discussion

Several in vitro investigations have shown the positive effects of copper on cells that regulate bone metabolism. Li and Yu [37] showed that copper ions can suppress the resorption of osteoclasts. Several investigators have confirmed that copper exerts positive effects in a dose-dependent manner. Low concentrations (0.1% *w*/*w*) of copper increased the viability and growth of osteoblasts, while higher concentrations (2.5 and 1% *w*/*w*) were shown to be toxic to cells [38]. In addition, the existence of copper stimulates the differentiation of MSCs toward the osteogenic lineage [39]. The copper content of femoral head necrosis tissues reported in previous studies varied. Milachowski [40] observed a decrease in copper levels in his study of the relationship between idiopathic femoral head ischemic necrosis and trace element metabolism. Y’mazaki’s [41] study showed the exact opposite results, with an increase in copper levels in subchondral bone and cartilage in ischemic femoral head necrosis. Therefore, we speculate that SONFH may be accompanied by an imbalance in copper metabolism. A study by Gonzalez-Reimers et al. [42] confirmed, through experiments in rats, that steroids increase muscle copper, iron, and zinc levels, as well as bone copper levels. In conclusion, we hypothesize that disorders of copper metabolism may be one of the mechanisms underlying the pathogenesis of SONFH. To test our hypothesis, we searched for SONFH-related and copper metabolism-related genes in public databases and identified PNP and SLC2A1 as DE-SONFHGs that are associated with copper metabolism through a series of analyses, such as GO and KEGG analyses. We also discovered a significant and the strongest negative correlation between PNP expression and plasmacytoid dendritic cell infiltration. The strongest negative correlations were found between PNP expression and plasmacytoid dendritic cell infiltration, between SLC2A1 expression and T follicular helper cell infiltration, between PNP expression and CD56 bright natural killer cell infiltration, and between SLC2A1 expression and T helper 17 cell infiltration. It was experimentally verified that both PNP and SLC2A1 were significantly downregulated in the peripheral blood of SONFH patients, as well as in the hormonal osteonecrosis samples, which was consistent with the results of our bioinformatics analysis.

Purine nucleoside phosphorylase (PNP) is a vital enzyme in purine metabolism. A missense SNP (rs1049564) in the PNP gene was found to be associated with high IFN levels in SLE. The rs1049564 T allele of PNP is a loss-of-function variant that triggers blockade of the S phase and activation of the IFN pathway in lymphocytes [43]. PNP deficiency induces apoptosis mediated by the p53 pathway in human pluripotent stem cell-derived neurons [44]. In our study, it was firstly determined that the down-regulation of PNP at the mRNA and protein levels resonated with the activation of the p53-mediated endogenous apoptotic signaling pathway in SONFH patients. We speculate that PNP downregulation in SONFH may mediate femoral head necrosis through the P53 pathway.

SLC2A1 is a gene that encodes a glucose transporter protein (GLUT1) that controls glucose uptake and is encoded on 1p34.2 [45]. This gene is essential for glucose metabolism, is involved in normal and tumor cell glycolysis, and can play a key role in the cell growth and proliferation of many tumor cells [46,47,48]. One study confirmed that in mice deficient in SLC2A1, the Wnt7b-induced bone anabolic function is blocked and bone formation is affected [49]. SLC2A1 (GLUT1) is one of the targets of miR-140-5p [50], and miR-140-5p may promote the development of femoral head necrosis through the ubiquitin proteasome system [51]. This study confirmed that SLC2A1 is downregulated in SONFH tissues as did the previous literature [52], suggesting that the biological process through which miR-140-5p targets SLC2A1 may be closely related to the development of SONFH.

Both PNP and SLC2A1 were enriched in signaling pathways associated with SONFH pathogenesis. The PNP gene was mainly enriched in “positive regulation of smoothened signaling pathway”, “positive regulation of cytokine production involved in immune response”, “glycosphingolipid biosynthesis lacto and neolacto series”, and “B-cell receptor signaling pathway”. SLC2A1 was mainly enriched in “cgmp metabolic process”, “vacuolar acidification”, and “glycosphingolipid biosynthesis lacto and neolacto series”. The main pathways clearly associated with SONFH in this study included “neutrophil extracellular trap formation”, “PI3k-Akt signaling pathway”, “mTOR signaling pathway”, “TNF signaling pathway”, and “HIF-1 signaling pathway”. Drug-induced glucocorticoid administration is considered to be a risk factor for femoral head necrosis. It has been proposed that glucocorticoid-induced platelet activation leads to disruptions in the local blood flow in the femoral head. Activated platelets can trigger neutrophil extracellular trap (NET) formation, leading to the ischemic necrosis of bone cells [53]. AKT/mTOR signaling pathway components are upregulated in a glucocorticoid-induced ONFH model, and human umbilical cord MSCs reduce macrophage polarization by inhibiting the AKT/mTOR signaling pathway and thus ameliorate necrosis and osteoclast apoptosis in a GC-induced model of ONFH [54]. In contrast to that in normal tissues, the expression of TNF-α in ONFH bone tissues was notably upregulated, and autophagy, apoptosis, and the p38 MAPK/NF-κB signaling pathway were significantly activated, suggesting a significant effect of the TNF signaling pathway in the pathogenesis of femoral head necrosis [55]. Some scholars have also confirmed the involvement of the HIF-1 signaling pathway in the pathogenesis of hormonal osteonecrosis. Animal experiments have confirmed that bone health supplements can expedite the formation of new bone, promote the resorption of damaged bone, inhibit the inflammatory response, and ultimately ameliorate SONFH through the HIF-1α/BNIP3 pathway [56].

Many studies have confirmed that copper plays a vital role in the function of the mammalian immune system; for example, animals with copper deficiency are more vulnerable to infections, whereas animals that consume an oral diet that is rich in copper are more resistant [57]; oral copper supplementation helps to maintain T-cell function in rats with acute spinal cord injury [58]. From the perspective of the immune system, the body needs balanced copper intake; moderate copper intake is sufficient for optimal immune function, while too much may be harmful to the organism [59]. Our study also found that the pathogenesis of SONFH is associated with multiple immune regulatory pathways. Therefore, we hypothesize that in the pathogenesis of SONFH, an imbalance in copper metabolism may cause disordered immune regulation in the body and thus induce femoral head necrosis. Recently, studies have shown that immune cell infiltration is associated with the progression of SONFH [35,60,61]. In the present study, we concluded that PNP expression had a notable and the strongest negative association with plasmacytoid dendritic cell infiltration, while PNP expression had a notable and the strongest positive association with CD56 bright natural killer cell infiltration. SLC2A1 had a notable and the strongest negative association with T follicular helper cell infiltration, and SLC2A1 had the strongest positive association with T helper 17 (Th17) cell infiltration. T follicular helper cells (Tfhs) are currently a research hotspot in basic immunology. Tfhs may contribute to the bone destruction that is associated with osteoporosis [62]. The balance between Treg and Th17-cell activity directly affects osteoclastogenesis and osteoblast/osteoblast coupling regulation [63]. In summary, immune cells may influence osteoblasts and osteoclasts in the pathogenesis of SONFH. Therefore, assessing the differences in the proportions of infiltrating immune cells in SONFH is valuable for elucidating the molecular mechanisms underlying SONFH and verifying molecular markers that are associated with immune infiltration.

Among the drugs predicted to target the key genes in this study and thus to affect SLC2A1, genistein (genistein) may be one of the most promising drugs. Genistein is an isoflavone that is also known as genistein, and it is an estrogen-like compound that is widely found in legumes [64]. It can prevent bone loss in human and rat models of osteoporosis directly by acting through estrogen receptors (ERs) on bone cells and indirectly by affecting thyroid follicular cell activity [65]. Phytoestrogens prevent methylprednisolone-induced femoral head necrosis and secondary osteoporosis in rats [66] The specific mechanism by which genistein ameliorates SONFH requires further investigation.

In the present study, we found notable differences in the expression levels of three copper death-related genes (DLD, PDHB, and MTF1) that are associated with PNP and SLC2A1 in SONFH. Dihydrothioctanamide dehydrogenase (DLD), which is also referred to as the E3 subunit of pyruvate dehydrogenase complex (PDHC) EC 1.6.4.3, is the third catalytic enzyme of PDHC, and it is a multifunctional mitochondrial matrix enzyme [67]. It has been demonstrated that DLD gene silencing prevents lipid peroxidation and iron-related death in vitro and in vivo [68]. Pyruvate dehydrogenase B (PDHB) encodes pyruvate dehydrogenase, which is a constituent enzyme of the pyruvate dehydrogenase multienzyme complex in mitochondria [69]. Recessive PDHB mutations cause pyruvate dehydrogenase complex (PDC) deficiency, which mainly affects the nervous system, such as developmental delays, seizures, and peripheral neuropathy [69]. Recent studies have found that DLD and PDHB positively regulate copper-related death [34]. Similarly, the current study revealed for the first time the high expression of DLD and PDHB in SONFH patients at the mRNA and protein levels, and we hypothesize that these gene alterations may disrupt pyruvate metabolism and thus induce the copper-related death of osteoblasts in the pathogenesis of SONFH. Further experimental studies are needed to elucidate the related mechanisms. Metal-regulated transcription factor 1 (MTF1) is a highly conserved zinc (Zn)-binding transcription factor in eukaryotes that responds to both metal overload and metal deficiency to protect cells from oxidative and hypoxic stress. A comparable Cu+-binding center was confirmed to be present in mammalian MTF1, suggesting that it may also respond to Cu [70]; however, in vitro experiments also confirmed that copper ions enhance MTF1 expression in myogenic cells [71]. The application of hormones can increase copper contents in muscle, bone, and even femurs [41,42]. Recent studies have confirmed that MTF1 is a negative regulator of copper-related death [34]. MTF-1 is one of the targets of miR-148-3p, which inversely regulates MTF-1 transcriptional activity [72]. Moreover, miR-148-3p is downregulated in bone marrow mesenchymal stem cells of mice with steroid-induced femoral head necrosis [73], which echoed the over-expression results of MTF in SONFH patients in this study. Thus, we speculate that low miR-148-3p might -increase the MTF-1 transcriptional level in the pathogenesis of SONFH. It is highly likely that MTF-1 is closely related to SONFH.

There were some limitations in the current study. For example, the number of peripheral blood samples and femoral head tissue samples is not large enough, and meanwhile, more in-depth functional analyses combining key genes with potential functions and processes, as well as vital copper death-related genes, are exigent for SONFH, which needs to be verified by collecting more clinical samples in the future.

## 5. Conclusions

In summary, this study offers new perspectives on the relationship between copper metabolism and SONFH. Steroids may target PNP and SLC2A1 to induce cuproptosis in osteoblasts and thus trigger femoral head necrosis; PNP and SLC2A1 may be potential drug targets and biomarkers for the diagnosis of SONFH. In the future, we will further study the specific mechanism by which these genes are involved in SONFH. In-depth studies on genistein as an effective agent for the treatment of hormonal femoral head necrosis should be continued. The present study also had the following limitations. First, the data used in this study were extracted from a single source, and the results may be biased to a certain extent. Second, functional experiments should be carried out to further elucidate the potential molecular mechanisms underlying hormonal femoral head necrosis.

## Figures and Tables

**Figure 1 biomedicines-11-00873-f001:**
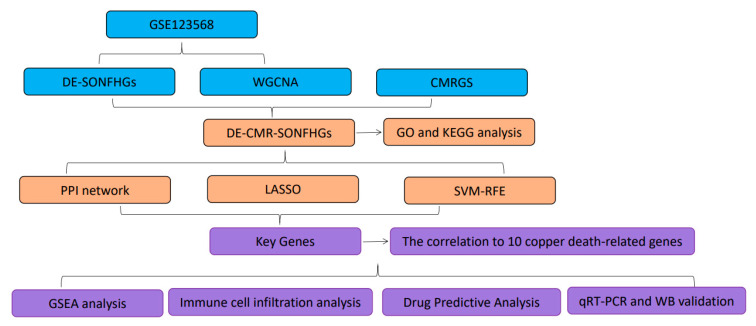
Workflow diagram clearly showing the major steps of this study.

**Figure 2 biomedicines-11-00873-f002:**
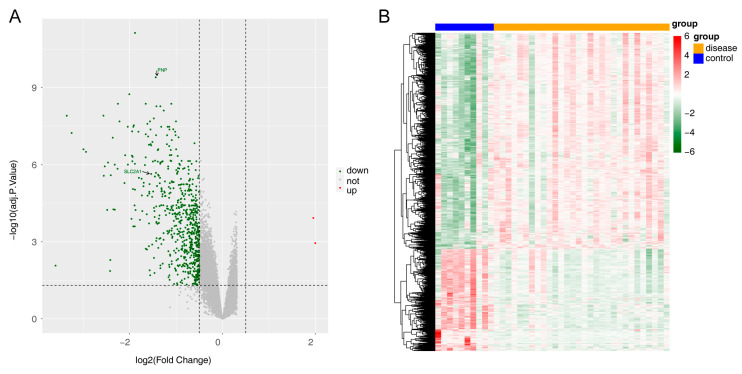
Identification of a total of 2036 differential expressed SONFH genes (DE-SONFHGs) between SONFH samples and control samples in the GSE123568 dataset. Volcano map (**A**) and heatmap (**B**) of 2036 DEGs.

**Figure 3 biomedicines-11-00873-f003:**
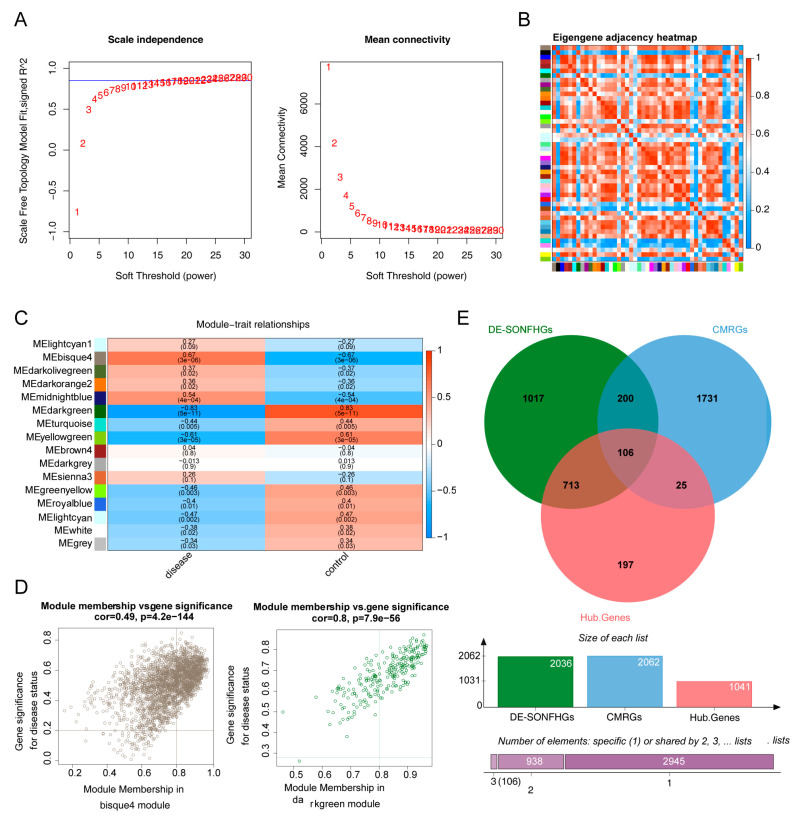
Identification of 106 differentially expressed CM-related SONFH genes (DE-CMR-SONFHGs) through the Weighted Gene Coexpression Network Analysis (WGCNA). (**A**) Analysis of the scale-free index for various soft-threshold powers (β); (**B**) HeatmapI of the association among identified modules. (**C**) Heatmap of the correlations between modules and clinical traits (disease and control). (**D**) Scatter plots of correlation between module members (MMs) and gene significance (GS) in two key modules. (**E**) Venn diagram of a total of 106 identified DE-CMR-SONFHGs.

**Figure 4 biomedicines-11-00873-f004:**
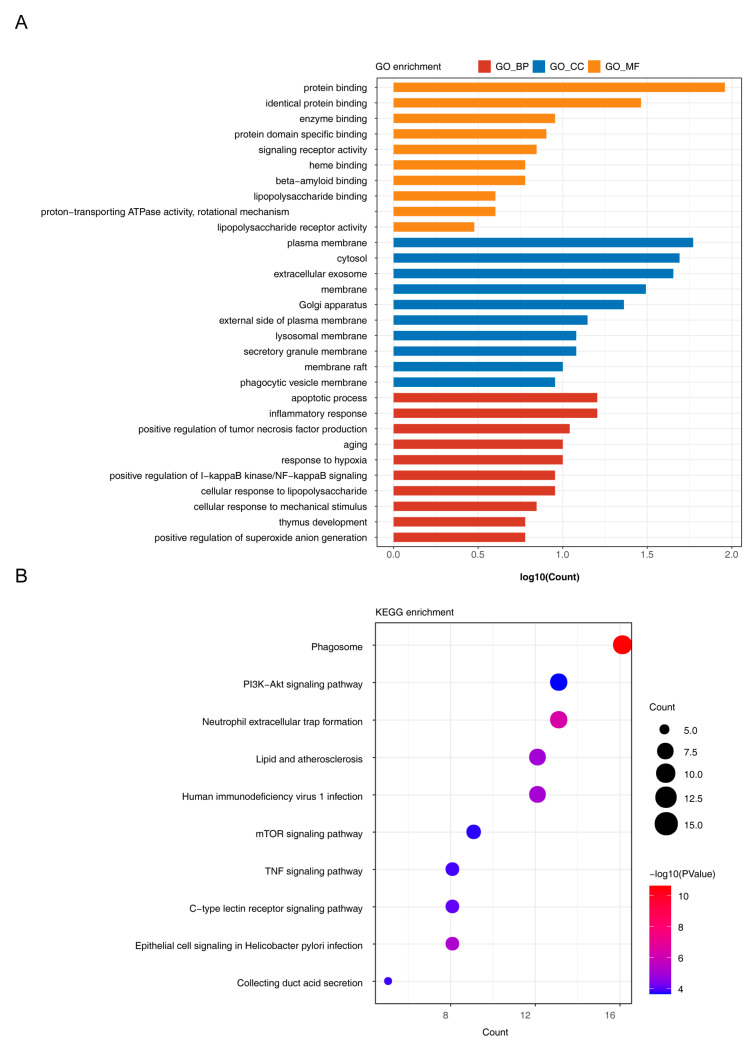
Gene Ontology (GO) and Kyoto Encyclopedia of Genes and Genomes (KEGG) enrichment of 106 DE-CMR-SONFHGs (**A**,**B**).

**Figure 5 biomedicines-11-00873-f005:**
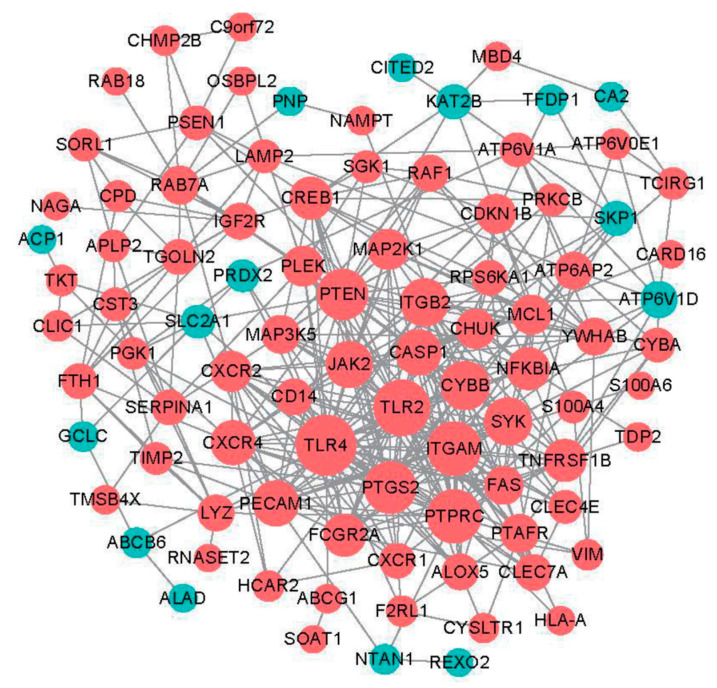
A protein–protein interaction (PPI) network consisting of 106 DE-CMR-SONFHGs was constructed. Red circles represent up-regulated genes, and green circles represent down-regulated genes.

**Figure 6 biomedicines-11-00873-f006:**
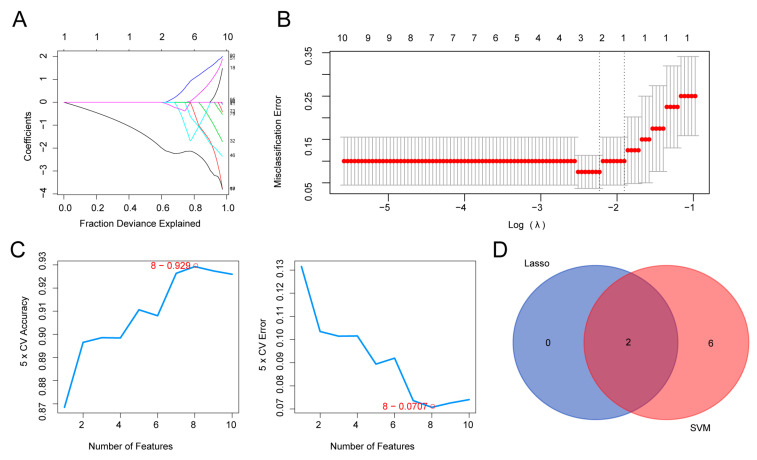
Least-Absolute Shrinkage and Selection Operator (LASSO) logistic regression and the Support Vector Machine (SVM) algorithm were utilized to screen key genes. (**A**,**B**) PNP and SLC2A1 were identified with LASSO; (**C**) RNASET2, PNP, SLC2A1, REXO2, CYBA, SOAT1, TFDP1, and LYZ were identified using SVM; (**D**) PNP and SLC2A1 were identified in the results of both LASSO and SVM.

**Figure 7 biomedicines-11-00873-f007:**
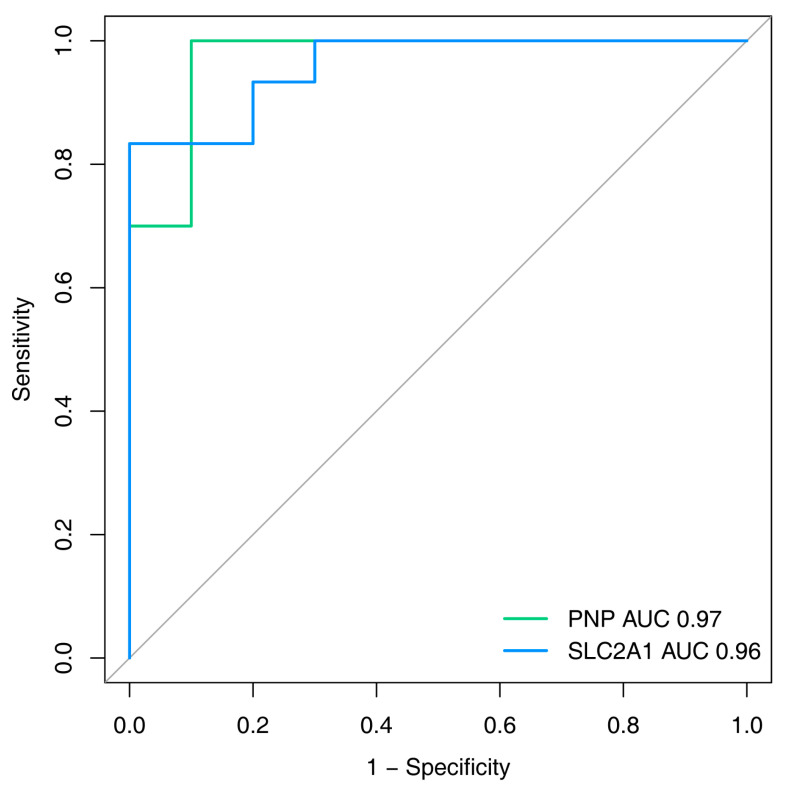
Receiver operating characteristic (ROC) curves of PNP and SLC2A1 in the GSE123568 dataset. The area under the curve (AUC) value of PNP was 0.97, and the AUC value of SLC2A1 was 0.96.

**Figure 8 biomedicines-11-00873-f008:**
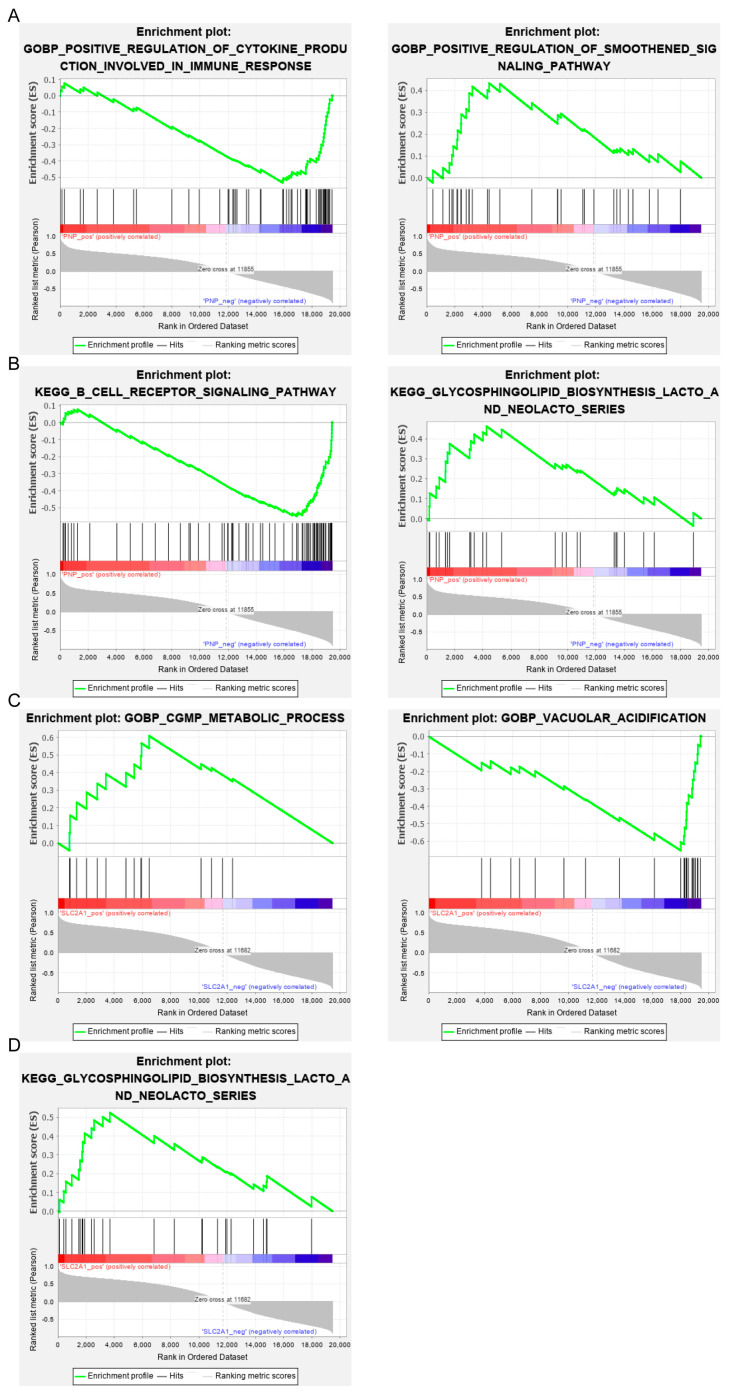
Single-gene GSEA of PNP and SLC2A1. (**A**,**B**) GO-BP and KEGG enrichment of PNP. (**C**,**D**) GO-BP and KEGG enrichment of SLC2A1. The threshold for enrichment significance was a NOM *p* value < 0.05. A positive enrichment score (ES) indicates that the tested gene set is enriched at the top of the pre-defined gene set, and a negative ES indicates that the tested gene set is enriched at the bottom of the pre-defined gene set. The ranked list matrix exhibits the correlation of the key gene with all other genes separately in the gene sets.

**Figure 9 biomedicines-11-00873-f009:**
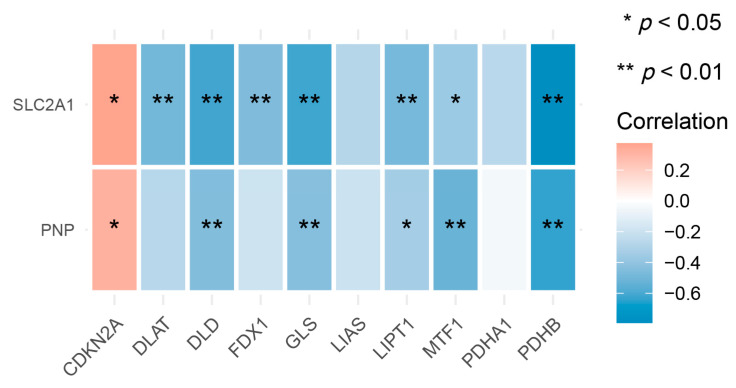
Heatmap of Pearson correlations between key genes and copper-death related genes; red represents a positive correlation, and blue represents a negative correlation.

**Figure 10 biomedicines-11-00873-f010:**
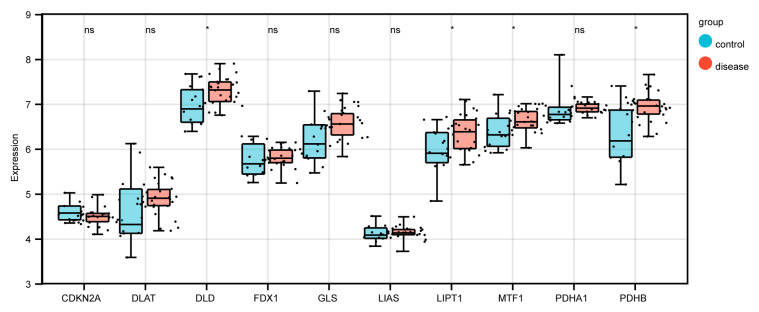
Boxplot of copper death-related gene expression between the disease samples and control samples (* *p <* 0.05).

**Figure 11 biomedicines-11-00873-f011:**
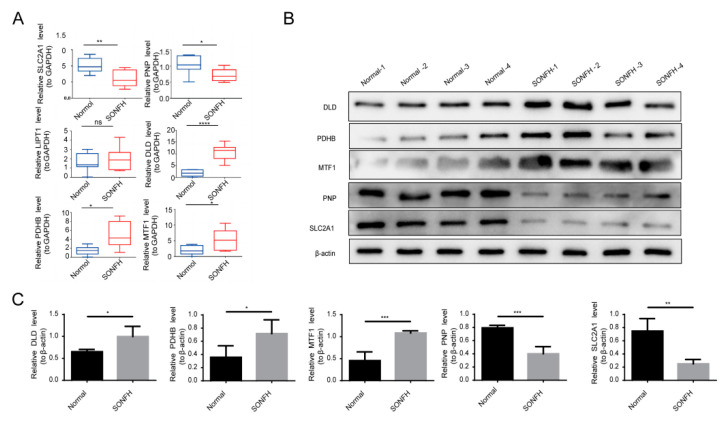
Expression vavification of the key genes, as well as the vital copper death-related genes. (**A**). PNP, SLC2A1, LIPT1, DLD, PDHB, and MTF1 expression levels were verified via quantitative real-time polymerase chain reaction (qRT–PCR) (n = 13). (**B**,**C**). DLD, PDHB, MTF1, PNP, and SLC2A1 expression levels were verified via Western blot (WB) analysis with quantification (n = 8), indicating the expression of DLD, PDHB, and MTF1 was notably increased in the SONFH group; the expression of PNP and SLC2A1 was notably reduced in the SONFH group. ns, no significant difference, * *p <* 0.05, ** *p <* 0.01, *** *p <* 0.001,**** *p* < 0.0001.

**Table 1 biomedicines-11-00873-t001:** Specific sequences of the primers used in the polymerase chain reaction (PCR) assay.

Gene	Forward Primers (5′-3′)	Reverse Primers (5′-3′)
GAPDH	CCCATCACCATCTTCCAGG	CATCACGCCACAGTTTCCC
SLC2A1	CTCATCAACCGCAACGA	AGTATGGCACAACCCGC
PNP	TCTCACACTAAGCACCGAC	ACCTCCTAATCCAGAACCA
LIPT1	GGAGAAGAAGTGGAGGAGGA	CCGTTGGGTTTATTAGGTGA
DLD	GCCGACGACCCTTTACTA	GCCTTCATCCTCTGCTTT
MTF1	TAATAATCCCACAATAACCAT	TAAAAAACACCTTCTCAACTT
PDHB	TGGAAAAGCCAAAATAGAAAG	CATAAGGCATAGGGACATCAG

**Table 2 biomedicines-11-00873-t002:** Eight candidate genes (RNASET2, PNP, SLC2A1, REXO2, CYBA, SOAT1, TFDP1, and LYZ) screened via Support Vector Machine (SVM).

Feature Name	Feature ID	Avg Rank
RNASET2	80	6.6
PNP	67	7
SLC2A1	87	9.6
REXO2	79	10.2
CYBA	32	12
SOAT1	89	12
TFDP1	19	13.4
LYZ	51	14

## Data Availability

The datasets used and/or analyzed during the current study are available from the corresponding author upon reasonable request.

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
