# Peer review of "Bioinformatics-Based Analysis of Key Genes in Steroid-Induced Osteonecrosis of the Femoral Head That Are Associated with Copper Metabolism"

_biomedicines, 2023, doi:10.3390/biomedicines11030873_

Round 1
Reviewer 1 Report
Comments:
The present original study applied combined bioinformatic methods to assess the key genes likely involved in steroid-induced osteonecrosis of the femoral head associated with copper metabolism.
The manuscript is well-written and readable.
The introduction, as well as the Discussion sections, are well organized.
The materials and Methods section is well-structured. Methods are clearly described, as well as study Results.
Concerns and Suggestions:
Abstract:
Please, follow the Journal’s guidelines (number of words, headings, etc.).
M&M:
Methods employed have been widely described. Did you compute the sample size to determine the number (13) of peripheral blood samples?
Why did you choose to compare 6 vs 7 samples and 4 hormonal osteonecrosis vs. 4 normal femoral head bone tissue?
Discussion:
Please, rephrase “Our analysis may be related to the imbalance between the 376 two investigators' subjects at baseline.”.
Reviewer 2 Report
In this study, Qi et al. aimed to understand relationship between copper metabolism and steroid-induced osteonecrosis of the femoral head (SONFH), and performed bioinformatics-based identification of copper metabolism associated genes involved in SONFH. They identified two genes (PNP and SLC2A) which had diagnostic value in distinguishing between SONFH and control samples. Additionally, the authors tried to identify copper death-related genes of which expression is either positively or negatively correlated with that of PNP and SLC2A, and identified several candidate genes. Finally, the authors confirmed that expression of PNP, SLC2A, and three copper death-related genes was differentiated between SONFH samples and control samples. The authors concluded that PNP and SLC2A could be potential targets of SONFH treatments and/or diagnostic biomarkers.
This is a good study. Appropriate utilization of bioinformatics to find candidate gene(s) which for real correlates in the disease. Although, as the authors pointed, the specific mechanisms by which these genes are involved in SONFH are yet unclear, this study will be one of cutting edges to understand the pathology and mechanisms underlying SONFH.
However, this study has several points that revision is required.
1. Single gene enrichment analysis: I don’t quite understand ‘single gene’ (not ‘single-sample’) GSEA analysis. How could be the analysis done? The authors need to explain more details of this analysis, and also cite references which utilize the same ‘single gene’ GSEA.
2. Immune cell infiltration analysis: the authors described the potential involvement of specific immune cell infiltration. However, given that they did not validate the results in SONFH patients, I suggest to bring the results in supplemental results.
3. Drug prediction analysis: the authors identified potential drugs which could target PNP and/or SLC2A. However, they did not validate effects of the drugs, and I suggest to bring the results in supplemental results.
4. Line 412-414: I didn’t find Table S2 and 6 in this paper.
5. Line 448-449: please cite references for this sentence.
Reviewer 3 Report
Comments to manuscript 2197482:
The authors used the following data base “SONFH-related dataset GSE123568” that already analyzed and published by Dr. Yanqiong Zhang in 2018 and already analyzed the differential expressed genes. I am not sure it is appropriate to reanalyze here by the authors although the databases are shared. It is kind of redundant work with suspicion of plagiarism. There is minimum original work in this study which repeated previous work’s results that will not contribute to the field. Authors should make clear why they used the others’ published data base as their main data source of this manuscript.
Following are some comments:
In many cases, authors did not give full name of the abbreviations or given after using many times of abbreviations.
|
Line 179 “The thermal procedure of cDNA synthesis” was wrong. It should be PCR. |
Figures legends lack of details and explanation.
Majority of the fonts in the figures are generated by software and lack of readability.
Figure 13, authors used both star and exact P value which is redundant.
Line 464,-465, “oestrogen” spelling wrong.
There is no discussion of the only work the authors did on the gene expression in their clinical samples.
Round 2
Reviewer 2 Report
The authors replied to my questions.
I have just one thing, please add the citation of the following studies (PMID: 36524127, PMID: 36237968) appropriately in the paper.
Reviewer 3 Report
The revised manuscript has improved compared to the first version. It is now acceptable for publishing in Biomedicine after correcting further minor errors:
Figure 5 legend both red and green representing up-regulated genes?
